# Non-IgE Mediated Hypersensitivity to Food Products or Food Intolerance—Problems of Appropriate Diagnostics

**DOI:** 10.3390/medicina57111245

**Published:** 2021-11-14

**Authors:** Dorota Myszkowska, Barbara Zapała, Małgorzata Bulanda, Ewa Czarnobilska

**Affiliations:** 1Department of Clinical and Environmental Allergology, Jagiellonian University Medical College, Botaniczna 3, 31-508 Kraków, Poland; malgorzata.l.lesniak@gmail.com (M.B.); ewa.czarnobilska@uj.edu.pl (E.C.); 2Department of Clinical Biochemistry, Jagiellonian University Medical College, Skawińska 8, 31-066 Kraków, Poland; barbara.zapala@uj.edu.pl

**Keywords:** adverse reactions to food, alternative tests, FLG genes

## Abstract

The diagnosis of food intolerance is tricky due to the different etiologies of adverse reactions. There is also a lack of clear rules for interpreting alternative tests used to diagnose these problems. The analyses of IgG4 concentration in serum or cytotoxic tests became the basis of elimination diets. However, it can result in nutritional deficiencies and loss of tolerance to eliminated foods. Our study aimed to assess the necessity of food elimination in four cases with food intolerance symptoms based on alternative diagnostic tests. Four patients without food allergies, who manifested diverse clinical symptoms after food, were presented due to the following factors: clinical history, diagnostic tests, elimination diet, and filaggrin gene (FLG) mutation. It was found that higher IgG4 levels against foods and higher cytotoxic test values are not clinically relevant in each of the studied individuals. They should not be decisive for the elimination of food products. The study of FLG-SNVs revealed the association of some clinical symptoms in patients with hypersensitivity to several food allergens and reported genetic variants in the FLG gene.

## 1. Introduction

Adverse reactions to food are becoming an ever-growing problem related to food hypersensitivity, including immunological reactions (both IgE-mediated and non-IgE-mediated) and non-immunological responses classified as “food intolerance” [1,2,3,4,5,6,7]. Various clinical symptoms manifest as skin, upper respiratory system, or most frequent digestive symptoms, often resembling allergic manifestations [1]. The variety of mechanisms of symptoms renders the diagnosis of adverse reactions to food tricky in clinical practice. They can also mimic the problems of chronic digestive conditions, such as irritable bowel syndrome (IBS) [8] or Crohn’s disease [9].

Often, neither the clinical history of the patient nor the results of diagnostic tests identify the food products responsible for the reported symptoms. When a non-immunological response after consuming a specific food is suspected, the definitive diagnosis based on documented clinical research does not propose a solution to the problem, which results in patients searching for unconventional, commercial diagnostic methods [6,10,11]. 

In vitro, ex vivo, and in vivo tests are proposed among the methods of non-confirmed efficiencies, which include so-called “alternative tests” [6,12]. Cytotoxic tests, which are based on the assessment of the changes in the number and size of leucocytes after the patient’s fresh blood incubation with a suspected allergen, are proposed as an alternative and safe tool for challenge tests; however, they are characterized by low repeatability and a lack of correlation with clinical symptoms [6,10,13,14]. It is stated that the presence of specific IgG antibodies against food allergens reflects the natural defense reactions of the body to allergen penetration due to the damage of the epithelial barrier and a specific humoral response to pathogens, being useful in the diagnosis of IBS [15] or migraine [16]. IgG4 subclass occurs in the lowest quantity, but in the case of food allergens, this subclass constitutes the higher amount of this class [17,18,19,20,21]. IgG4 occurrence indicates a repeatable exposure to high doses of food components, confirmed by studies on healthy individuals [6,10,12,21,22]. According to the World Allergy Organization (WAO) and allergy-related organizations’ recommendations, measuring IgG antibodies is helpful for confirming tolerance to food products [6,21].

In the studies on food intolerance, the mutations in the filaggrin gene (FLG) are considered important factors that predispose atopic diseases [23,24] that result primarily in epidermal barrier dysfunction and mucosal surface of gastrointestinal tracts, provoking the more effortless penetration of food proteins [25,26,27]. There has been a gap in understanding the possible role of the filaggrin gene defects, especially in non-IgE-mediated reactions to food. Our study aimed to assess the necessity of food elimination in patients with food intolerance symptoms based on alternative diagnostic tests. 

## 2. Material and Methods

Four selected patients (women, aged 23–49 years) with different pathologic symptoms of food intolerance and excluded an IgE-mediated food allergy (such as negative skin prick tests and IgE concentration for food allergens) were included in the assessment with the patient’s permission. The following tests were performed in all cases. (1) IgG4 concentrations for eggs, nuts, casein, peach, and omega-5-gliadin were measured sera using fluoroenzymeimmunoassay (FEIA) (Thermo Scientific, Phadia AB) and two commercial tests: (2) ImmuPro basic test based on immunoenzymatic reaction (ELISA) (Institute of Microecology, Warsaw, Poland) and (3) cytotoxic test, in which the damages in leucocytes after contact with food are estimated (Vimed, Warsaw). Within the examination of the alternative test, the company’s reference values were considered. For IgG4 measurements, the level of 0.35 UI/l, the same as for venom allergens, was taken into account according to the literature [28]. A four-week diagnostic elimination diet was recommended in the studied patients, including the less tolerated products and those of a higher IgG4 concentration, followed by introducing the eliminated foods one by one every two weeks. All the tested patients were under the supervision of a dietician, and they filled in patients’ diary cards by indicating any side effects during the products’ re-entry. 

Peripheral blood samples of all selected patients were collected to EDTA tubes at the same time as serum samples. Genomic DNA samples were extracted from the patients’ peripheral blood by using the QIAamp DNA Mini Kit (Qiagen, Hilden, Germany). As a control, DNA from healthy, unrelated individuals was used. Polymerase chain reaction (PCR) was used to amplify all the coding exons and adjacent intronic regions of FLG. The sequences of primers specific to the FLG gene were used according to Sandilands et al. [29]. Amplification was performed by using the FastStart PCR Master Kit (ROCHE Diagnostics). The PCR products were purified by using NucleoSpin Gel and PCR Clean-up (Macherey Nagel) and sequenced with the BigDye Terminator v.3.1 Cycle Sequencing Kit (Life Technologies, Foster City, CA, USA). The sequences were analyzed by using AB DNA Sequencing Analysis Software v.5.2. (Applied Biosystems). Furthermore, they were compared to the FLG gene reference (NCBI GeneBank Reference Sequence Accession Numbers: NM_002016.2; ENSEMBL database Accession Numbers: ENST00000368799.2). All detected FLG variants were analyzed by using Human Gene Mutation Database [30] and ClinVar [31].

## 3. Results—Case Reports

Case 1 was a 40-year-old woman with a history of food intolerance, preliminarily diagnosed as an unspecified allergy, and she was admitted in 2018. Gastrointestinal symptoms, such as cramping, abdominal pain and flatulence, and problems with defecation, manifested for six years, usually up to 30 min after eating. The cytotoxic test (Metabolic code 203) indicated, among others, gluten intolerance and a mandatory recommended elimination of products containing gluten. Due to the fact that celiac disease and gluten allergy were excluded, the introduction of small amounts of gluten products under the control of clinical symptoms was recommended before IgG4 examination. The highest value of IgG4 concentration was obtained for egg white and casein (Table 1). A four week diagnostic diet was proposed by excluding flour, dairy, and egg products and then successively introducing these products one by one every two weeks. After introducing eggs, no changes in the patient’s wellbeing were reported, while stomach pain was recorded after drinking milk. Moreover, when gluten products were reintroduced for two days only, digestive problems were noted (Table 2). Currently, the patient is on a strict gluten-free diet. Preliminary diagnosis was corrected into malabsorption induced by intolerance. Three heterozygous variants, c.2528G > C (rs558054487), c.8673G > T (rs57672167), and c.9313T > G (rs2065958), were reported in patient one (Figure 1A–C).

Case 2 was a 44-year-old woman diagnosed with an unspecified allergy due to non-specific symptoms after consuming food, observed for the past five years. The following problems were recorded: hand trembling (after eating almonds), feeling alternating cold and hot, stinging eyes, feeling “lumps in the throat” and on the cheeks, drooling, andscratchy throat (after cookies and biscuits). Two years before the visit, she underwent a cytotoxic test (Metabolic Code 203) and confirmed a higher level of intolerance to most products, mainly wheat, yeast, and dairy products. The patient noticed an improvement after eliminating some of the products but was concerned about a weight loss of 5 kg. The increased results of IgG4 concentration against food allergens were obtained for eggs and peach (Table 1), which were eliminated for four weeks, followed by their introduction one after the other. At present, the patient tolerates eggs in small quantities. However, she avoids peaches, especially raw fruits, which may provoke cross reactions with pollen allergens. The patients are sensitive (positive skin prick tests (SPT) for birch, hazel, and alder allergens and a periodic runny nose in spring). She rarely drinks milk, which is also suspected of exacerbating the symptoms. The elimination of bread was also recommended due to the clinical symptoms observed in the patient (Table 2). One heterozygous variant c.9536T > G (rs2065957) was detected in patient two (Figure 1D).

Case 3 was a 23-year-old woman with a suspected food allergy who has been observing gastrointestinal symptoms, including diarrhea and flatulence, and, less frequently, skin symptoms for about 2–3 years. As the symptoms occurred several hours after ingestion, it was not easy to observe what kind of products might have caused them. She connected the symptoms with dairy products and eliminated them for six months, bringing about short-term improvement. Therefore, she rarely eats dairy products. She also attempted to eliminate eggs from her diet, but after reintroducing them, she did not observe any difference in symptoms manifestation, although she did not eat scrambled eggs. She thought that the symptoms could also have been intensified by wheat bread, citrus fruits, or cocoa. An ImuPro test (in IgG class) showed a level of IgG ≥ 20 µg/mL (defined as “high”) for some vegetables, such as broccoli, red cabbage, carrots, celery, and for eggs and yeast, which were eliminated together with eggs. The patient is very fond of egg dishes and, therefore, did not fully comply with these recommendations. Due to abdominal discomfort, she limited the consumption of bread and dairy products. The highest IgG4 titer was obtained for egg white and casein; thus, a 4-week diagnostic elimination diet was introduced, excluding dairy and egg products (Table 1). Baked and boiled eggs are well tolerated. However, flatulence still appears after the consumption of dairy products (cottage cheese and yogurt), although cheese was tolerated well (Table 2). The introduction of low-lactose products (lactose-free cheese is tolerated), testing for lactose intolerance, and a visit to the gastroenterologist were recommended. In patient three, two heterozygous variants were identified as follows: c.9313T > G (rs2065958) and c.9536T > G (rs2065957) (Figure 1E,F).

Case 4 was a 49-year-old woman suspected of food allergy who came to CEAC with abdominal ailments, which appeared for the first time in 2012 after eating Russian dumplings and an initial diagnosis of unspecified allergy. The patient reports that abdominal pain frequently appeared after eating fatty foods, dairy products, eggs, and sugar. However, it is difficult to state them accurately because they usually appear 2–3 days after consumption. The abdominal ultrasound examination performed in 2014 revealed no pathological changes, while endoscopy of the upper part of the gastrointestinal tract (in 2015) singled out the presence of a sliding hernia of the esophageal solution. An ImuPro test for food intolerance showed positive results for wheat, rye, barley, oats, gluten, yeast, banana, and pineapple (Table 1). Discontinuation was recommended as follows: products containing gluten proteins, avoiding yeast, sugar, alcohol, and consumption of lactose-free products. Serological and pathomorphological diagnostics towards celiac disease were negative. The IgG4 examination indicated the highest value for eggs and peaches (Table 1). An observational elimination diet was recommended for dairy products (because of symptoms occurrence), eggs, bread (mild symptoms), and peaches, after which she did not report any severe symptoms after consuming dairy products (Table 2). After consuming dumplings and using some cosmetics, abdominal discomfort and skin symptoms (erythema on the forehead) were noted. Comparable to patient three, the same variants (c.9313T > G (rs2065958) and c.9536T > G (rs2065957)) were reported in patient four (Figure 1G,H).

## 4. Discussion

Currently, unconventional diagnostic methods and laboratory tests confirm food allergy or intolerance [6,12]. However, the studies indicating their clinical usefulness are lacking, and they are treated as non-reproducible and do not correctly assess sensitivity and specificity. As qualitative or sub-quantitate procedures, cytotoxic tests are not recommended for diagnostics of food adverse reactions as their reproducibility and correlations with clinical symptoms were insufficient [9,12,32]. In the cases 1 and 2, who underwent cytotoxic tests, the results should have been interpreted slightly differently. Milk products elimination based on cytotoxic tests was inadequate for the test results because the higher value (+3) in case 2 did not indicate an elimination diet.

In contrast, in case 1, the avoidance of milk products was not necessary (+1). Low IgG4 values could have been related to milk intolerance; thus, she was told to test different products in small quantities instead of using a restrictive diet. Weight loss and nutritional deficiencies may occur in patients who eliminate a lot of food by themselves without a short-term (1–4 weeks) testing diet [33]. Eggs were well tolerated by case 1, while in the case 2, a lower IgG4 value could have been related to the slight symptoms. An elimination diet was not ordered according to the cytotoxic test (+2), although many egg dishes were not well tolerated.

In spite of “higher values” in cytotoxic tests for wheat allergens and low IgG4 concentrations, cereal products were eliminated, and after their reintroduction, the values maintained the same levels. Considering the low IgG4 value, this could be explained as a problem of gluten intolerance, called “non-celiac gluten sensitivity” (NCGS), and a gluten-free diet is the only effective treatment [34].

No objective studies indicating the level of sensitivity and specify of IgG tests were performed except in observational case-control studies on the group of 319 Hashimoto patients, who demonstrated higher (>30 U/l) IgG concentrations against many food allergens. However, in the control group of healthy individuals, increased IgG levels against wheat, eggs, and cow milk were obtained as follows: 91%, 82%, and 73% simultaneously [35]. The study performed in a vast group of 21,305 adult participants from different regions of China showed a high variability of IgG levels both in healthy and in symptomatic Chinese adults [36]. Jansen et al. did not find any differences between IBD patients and controls, and their levels did not correlate with food intolerance. In turn, Zuo et al. [37] reported that serum IgG antibody titers relative to some common foods increased in IBS and FD patients compared to controls. However, the clinical significance of these results was not confirmed. The other problem with IgG tests is the lack of a fixed range of reference values. Commercial tests indicate the value of 7.5 µg/L IgG concentration against food allergens as the higher value named as class 1, which is followed by the following three classes (up to > 50 µg/L). In the group of 130 healthy adults tested by Martins et al. [38], a range from 2 µg/L to 88.9 µg/L against 33 different foods was obtained. The authors concluded that the reference intervals should benefit clinicians in interpreting serum IgG allergen results, but in some instances it is separately applicable in cases only concerning a given food.

In our study, values above 7.5 µg/L of IgG were obtained for eggs and wheat, while the same values were observed in average persons (2–30.2 µg/L and 2–60.3 µg/L, respectively, according to Martins et al. [38]). In the case of patient three, the elimination of eggs was introduced based on a commercial examination, while the IgG4 level indicated a higher tolerance of these products which can be eaten without any restrictions. Higher IgG levels against wheat were obtained in case 4 after the elimination of all grain products, while IgG4 concentration was deficient (0.01 µg/L), suggesting non-celiac gluten intolerance. Increased IgG values in the ImuPro test and a slight increase in IgG4 concentration and the occurrence of the symptoms may have suggested milk product elimination. However, no indications after unconventional diagnostics were ordered. The patient should have tested lactose-free products in order to minimize a risk of any adverse reactions [39]. Similarly, in case 4, milk product elimination was not performed before IgG4 testing.

FLG genotyping was an exciting part of this study. FLG encodes a large protein called pro-filaggrin, which has been reported to play a critical role in maintaining an effective skin barrier against the environment [40]. In patients one, three, and four, we identified rs2065958 and rs2065957 variants, previously reported by Kim et al. [41], mainly in patients with higher reactivity to milk, soya bean, wheat flour, pork, and mite allergens related to non-specific hand or foot dermatitis and scalp scale. Patients 3 and 4 presented gastrointestinal and skin symptoms and were tested to eliminate dairy products, flour, and eggs from their diet. The synonymous variant rs57672167, identified only in patient one, was reported previously in patients with ichthyosis [42]. In the same patient, the rs558054487 variant was identified and was of unknown significance. The association between FLG-SNVs and observed food intolerance symptoms strongly suggested that FLG may be involved in pathomechanisms of food hypersensitivity. It could be supposed that the weakened protective function of filaggrin may increase the permeability of the gastrointestinal mucosa, resulting in different problems after consumption. To the best of our knowledge, few reports show the correlations between FLG-SNVs and food intolerance; thus, we introduce an additional viewpoint by suggesting the critical role of FLG in food-related problems, which should be examined in more detail [43].

## 5. Conclusions

There is currently no cure for food intolerances, and the best method to avoid symptoms is to avoid certain foods or to eat them less often and in smaller amounts. The standard for food intolerance testing is a temporary elimination diet followed by a controlled food challenge because no diagnostic tests are recommended for confirming non-allergic adverse reactions to food. The estimation of the induced tolerance to foods based on IgG4 measurement seems non-objective due to there being no compliance between the patients’ reaction and the IgG4 level. An effective diagnosis should be conducted by using the following: (i) clinical history, which may help identify the role of diet or other factors that worsen symptoms, and (ii) testing short-term elimination diet followed by food reintroduction performed under the supervision of a dietitian. Prolonged restricted diets can result in problems with adequate nutrition. The study of FLG-SNVs revealed the association of some clinical symptoms in patients with hypersensitivity to several food allergens and reported genetic variants in the FLG gene.

## Figures and Tables

**Figure 1 medicina-57-01245-f001:**
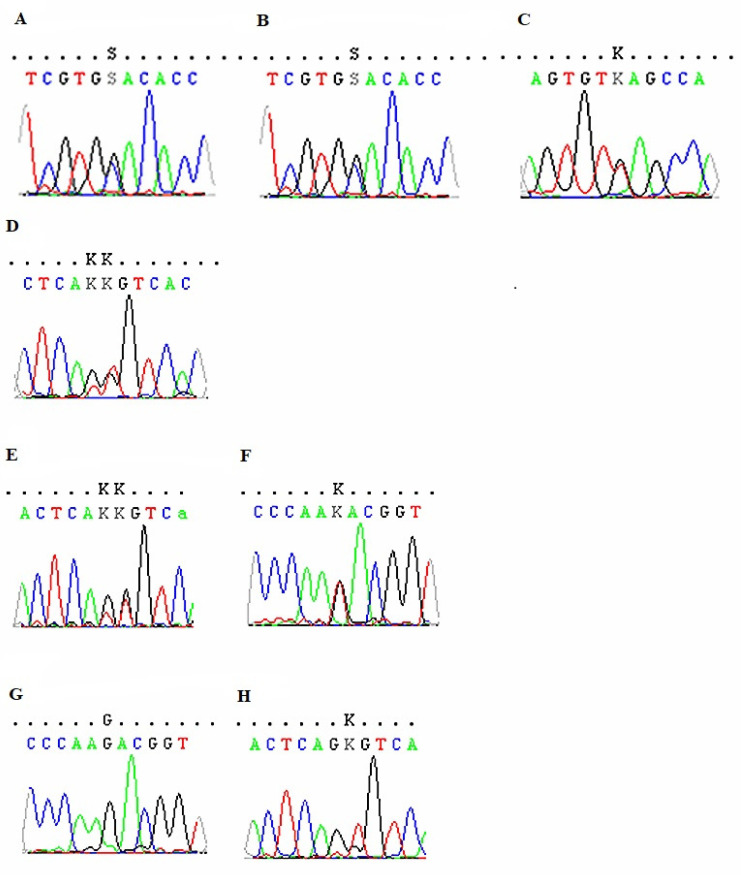
FLG sequencing electropherograms of the four (1–4) studied cases—variants in case 1: (**A**)—[p.(Gly843Ala)]; (**B**) —[p.(Val2891=)]; (**C**)—[p.(Tyr3105Asp)]; variant in case 2: (**D**)—p.(Val3179Gly)]; variants in case 3: (**E**)—[p.(Tyr3105Asp)]; (**F**)—[p.(Val3179Gly)]; var-iants in case 4: (**G**)—[p.(Tyr3105Asp)]; (**H**)—[p.(Val3179Gly)].

**Table 1 medicina-57-01245-t001:** Results of IgG4 concentration measurements and food intolerance tests (cytotoxic test and ImuPro test).

Kind of Food	Case 1		Case 2		Case 3		Case 4	
	Cytotoxic Test (Metabolic Code 203)	IgG4 (mgA/l)	Cytotoxic Test (Metabolic Code 203)	IgG4 (mgA/l)	Test ImuPro	IgG4 (mgA/l)	Test ImuPro	IgG4 (mgA/l)
Peach	0	2.68	0	4.1	No test	0.51	No test	1.53
Egg	0	23.1	+2	3.41	Extremely high >20 mg IgG/ml	5.46	+1 (7.54–12.49 mg IgG/mL)	1.44
Cow milk (casein *)	+1	1.26	+3	0.81	Increased >7.5 mg IgG/ml	1.49	Not increased	0.36
Peanuts	No test; +2 (walnuts), +3 (hazel)	0.05	+3 +2 (walnuts)	0.03	No test	1.07	+2 (12.5–19.9 mg IgG/mL)	0.01
Grains (omega-5-gliadine *)	+3 (wheat)	0.01	+2 (wheat)	0.01	Only wheat-increased	0.03	+3 (gluten) (20–49.9 mg IgG/mL) +3 (wheat) (20–49.9 mg IgG/mL)	0.01
Fish	+1/+3 (different genera)	0.00	+1/+3 (different genera)	0.01	No test	0.01	+1 (plaice)	0.00

* Specific allergen to which IgG4 concentration was measured. The results of cytotoxic and ImuPro tests were given adequately to the original description.

**Table 2 medicina-57-01245-t002:** Assessment of symptoms severity after food consumption before elimination and after reintroduction the products. Products in bold were eliminated after IgG4 concentration measurement (value above 0.1 mgA / L) and after symptoms assessment. OAS (Oral Allergy Syndrome).

Symptoms Severity after Food Consumption before Elimination and after Products Reintroduction								
	Case 1		Case 2		Case 3		Case 4	
Food Products (Allergens *)	Before Elimination	After Reintroduction	Before Elimination	After Reintroduction	Before Elimination	After Reintroduction	Before Elimination	After Reintroduction
Peach	No symptoms	No elimination	**No symptoms**	**No symptoms (OAS possible)**	No symptoms	No elimination	**No symptoms**	**No symptoms**
Egg	**No symptoms**	**No symptoms**	**No symptoms**	**No symptoms**	**Symptoms**	**Symptoms (slight)**	**No symptoms**	**No symptoms**
Cow milk (casein *)	**Symptoms**	**Symptoms**	Symptoms	No elimination	**Symptoms**	**Symptoms**	**Symptoms**	**Symptoms (slight)**
Peanuts	No symptoms	No elimination	No symptoms	No elimination	No symptoms	No elimination	No symptoms	No elimination
Grains (omega-5-gliadine *)	**Symptoms**	**Symptoms**	**Symptoms**	**Symptoms**	Symptoms	No elimination	**Symptoms**	**Symptoms (slight)**
Fish	No symptoms	No elimination	No symptoms	No elimination	No symptoms	No elimination	No symptoms	No elimination

* The concentration of a specific allergen component was measured. Bold was used to indicate products in which were eliminated.

## Data Availability

Data presented in this study were collected from the medical database of the University Hospital in Kraków with the consent of hospital management. The operational hospital system secures this database, and it is not possible to access the data externally.

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
