# Peer review of "Non-IgE Mediated Hypersensitivity to Food Products or Food Intolerance—Problems of Appropriate Diagnostics"

_medicina, 2021, doi:10.3390/medicina57111245_

Round 1

Reviewer 1 Report

After reviewing this manuscript, I have one major concern. The article needs to be reorganized. After reading this manuscript, I am confused about the authors’ findings and conclusions. One of the reasons is that there are many grammar mistakes, and some sentences are too long to read. In addition, the authors stated each case but did not summarize their similarities and differences. The relationship among cytotoxic test, IgG4, Test ImuPro and gene variants was not discussed. Second,

L27: The introduction needs to be reorganized. Starting from L37, the authors described many details related to IgE-mediated food allergy. While the purpose of this study is to investigate the filaggrin gene in non-IgE-mediated food allergies.

L49: Change to italic.

L74-76: Rewrite this sentence. Do the authors mean those patients have a food intolerance but not an IgE-mediated food allergy?

L79: What are alternative tests?

L83-85: Rewrite this sentence; it is very confusing. What are less tolerated products? What do the authors mean products of a higher IgG4 concentration? What are eliminated foods?

L88-89: When to collect the DNA samples? The collection period should be very critical to this study.

L105-107: Rewrite. This is not a finished sentence, and I could not understand it.

L114: The digestion problem from which food appeared?

What is beach food?

Author Response

To the Reviewers of the Medicina

Authors: We would like to thank the Reviewers for their comments, especially we thank the Reviewers for identifying and articulating potential weaknesses that we could specifically address to improve the re-submission. We have addressed these concerns individually below and accordingly made relevant changes to improve the submission. Critical key points that were revised

Comments for Reviewer #1:

Comments and Suggestions for Authors

After reviewing this manuscript, I have one major concern. The article needs to be reorganized. After reading this manuscript, I am confused about the authors’ findings and conclusions. One of the reasons is that there are many grammar mistakes, and some sentences are too long to read. In addition, the authors stated each case but did not summarize their similarities and differences. The relationship among cytotoxic test, IgG4, Test ImuPro and gene variants was not discussed.

Authors: English was corrected, sentences were shortened, and the text was reorganized. The evident relationship among the results of the presented tests was not stressed, because our intension was to indicate that the elimination diet should not be based on one of them only. As it was presented, the results changed regarding the food and the patient condition. Only four individuals were studied, so it is insufficient material for the comparative analyses. We are collecting the data obtained from the higher group of patients to try to find an evident statistical relation among the results. It was stressed that two of the identified variants: rs2065958 and rs2065957, previously reported as responsible for foot dermatitis, mainly in patients with higher reactivity to milk, soya bean, wheat flour, pork, and mite allergens, could also cause the damages in the gastrointestinal mucosa.

Second,

L27: The introduction needs to be reorganized. Starting from L37, the authors described many details related to IgE-mediated food allergy. While the purpose of this study is to investigate the filaggrin gene in non-IgE-mediated food allergies.

Authors: The indicated part of the text was shorten and the detailed information on the IgE-mediated reactions diagnostics was mentioned only. The aim of the study was to assess the necessity of food elimination in patients with food adverse symptoms based on alternative diagnostic tests. The defects in filaggrin genes were described as a novel approach in the food intolerance and/or non-IgE-mediated hypersensitivity. 

L49: Change to italic.

Authors: Was changed.

L74-76: Rewrite this sentence. Do the authors mean those patients have a food intolerance but not an IgE-mediated food allergy?

Authors: The presented problem was clarified to make it clearer.

L79: What are alternative tests?

Authors: The names of the commercial tests, made by the patients were added. The term “alternative” was replaced by “commercial”. Some of the tests, not recommended to be used in the diagnostics of non-IgE-mediated allergy or food intolerance are called “alternative”.

L83-85: Rewrite this sentence; it is very confusing. What are less tolerated products? What do the authors mean products of a higher IgG4 concentration? What are eliminated foods?

Authors: The sentence was rewrite.

L88-89: When to collect the DNA samples? The collection period should be very critical to this study.

Authors: Peripheral blood samples of all selected patients were collected to EDTA tubes at the same time as serum samples. The patients were asked not to eliminate the products provoking the adverse symptoms before blood samples collection.

L105-107: Rewrite. This is not a finished sentence, and I could not understand it.

Authors: The sentence was corrected.

L114: The digestion problem from which food appeared?

Authors: The digestive problem referred to the gluten products, reintroduced after elimination diet. The sentence was divided into two parts, to be more clear.

What is beach food?

Authors: Many sorry for this grammar mistake. It should have been “peach”. The error was corrected.

Reviewer 2 Report

Dear Editor,

you have asked me to review the manuscript with the title "Non-IgE mediated hypersensitivity to food products or food intolerance – problems of appropriate diagnostics"

The manuscript is interesting to read, it outlines the current challenges in the diagnosis of food intolerance. Some of the diaganostic tests, that were highlighted in the manuscript, are IgG4 levels in the serum, cytotoxic tests, and mutations of FLG gene.

My first general comment: The written scientific language is sometimes not understandable and the manuscript contains long sentences and non-scientific terms in some places, e.g., the term “more and more” is not scientific.

Other comments:

  • In the introduction-page 2: In vitro, ex vivo and in vivo ….. methods. This sentence is not complete. Alternative methods of what?
  • In the introduction-page 2: Among IgG subclasses, IgG4 occurs ……. intolerance (6,24). The authors stated contradictory facts here, so please reform this paragraph, by stating that IgG is either beneficial in the diagnosis of food intolerance or not, or separate the two contradictory facts using “ in contrast, on the other hand, on the contrary, etc.”
  • Material and Methods- page 2: Please divide the methods into parts “method 1, 2, 3” and provide the suppliers locations (company, city, state, country) for all of the material.
  • Results-page 3: Please describe shortly that you will mention four cases in upcoming results, please use the word “case 1, 2, 3, 4” instead of the word "patient", e.g., in case 3, the patient suffered from…., etc.
  • Results-page 3: Please provide high quality graphics “600 pixel”. Change the numbering of the subfigures into Latin, e.g., A “ I, II, II”, B “ I, II”, etc. And also in the legend, there is no 1,2,3,4 in the whole figure! Please describe only what exists in the figure.

Author Response

To the Reviewers of the Medicina

Authors: We would like to thank the Reviewers for their comments, especially we thank the Reviewers for identifying and articulating potential weaknesses that we could specifically address to improve the re-submission. We have addressed these concerns individually below and accordingly made relevant changes to improve the submission. Critical key points that were revised

Comments for Reviewer #2:

Comments and Suggestions for Authors

The manuscript is interesting to read, it outlines the current challenges in the diagnosis of food intolerance. Some of the diaganostic tests, that were highlighted in the manuscript, are IgG4 levels in the serum, cytotoxic tests, and mutations of FLG gene.

My first general comment: The written scientific language is sometimes not understandable and the manuscript contains long sentences and non-scientific terms in some places, e.g., the term “more and more” is not scientific.

Authors: Think you for this remark, the correction was done in the whole text.

In the introduction-page 2: In vitro, ex vivo and in vivo ….. methods. This sentence is not complete. Alternative methods of what?

Authors: The sentence was corrected.

In the introduction-page 2: Among IgG subclasses, IgG4 occurs ……. intolerance (6,24). The authors stated contradictory facts here, so please reform this paragraph, by stating that IgG is either beneficial in the diagnosis of food intolerance or not, or separate the two contradictory facts using “ in contrast, on the other hand, on the contrary, etc.”

Authors: The sentence was corrected.

Material and Methods- page 2: Please divide the methods into parts “method 1, 2, 3” and provide the suppliers locations (company, city, state, country) for all of the material.

Authors: The additional details were added. I would like to stress, that IgG4 tests were performed in frame of the project, in the recommended medical laboratory of the Centre of Allergology, University Hospital in Kraków, while commercial tests were made by the patients by themselves.

Results-page 3: Please describe shortly that you will mention four cases in upcoming results, please use the word “case 1, 2, 3, 4” instead of the word "patient", e.g., in case 3, the patient suffered from…., etc.

Authors: Thank you for this good point, the text was corrected.

Results-page 3: Please provide high quality graphics “600 pixel”. Change the numbering of the subfigures into Latin, e.g., A “ I, II, II”, B “ I, II”, etc. And also in the legend, there is no 1,2,3,4 in the whole figure! Please describe only what exists in the figure.

Authors: Fig. 1 was corrected.

Authors: We hope that I answered properly for all your comments.

Thank you for your re-consideration of this manuscript.

Round 2

Reviewer 1 Report

All the previous comments have been addressed by the authors.